# Juvenile Amyotrophic Lateral Sclerosis: A Review

**DOI:** 10.3390/genes12121935

**Published:** 2021-11-30

**Authors:** Tanya Lehky, Christopher Grunseich

**Affiliations:** 1EMG Section, National Institute of Neurological Disorders and Stroke, National Institutes of Health, Bethesda, MD 20892, USA; 2Neurogenetics Branch, National Institute of Neurological Disorders and Stroke, National Institutes of Health, Bethesda, MD 20892, USA; Christopher.Grunseich@nih.gov

**Keywords:** motor neuron disease, familial amyotrophic lateral sclerosis, *FUS*, *SETX*, *ALS2*

## Abstract

Juvenile amyotrophic lateral sclerosis (JALS) is a rare group of motor neuron disorders with gene association in 40% of cases. JALS is defined as onset before age 25. We conducted a literature review of JALS and gene mutations associated with JALS. Results of the literature review show that the most common gene mutations associated with JALS are *FUS*, *SETX*, and *ALS2*. In familial cases, the gene mutations are mostly inherited in an autosomal recessive pattern and mutations in *SETX* are inherited in an autosomal dominant fashion. Disease prognosis varies from rapidly progressive to an indolent course. Distinct clinical features may emerge with specific gene mutations in addition to the clinical finding of combined upper and lower motor neuron degeneration. In conclusion, patients presenting with combined upper and lower motor neuron disorders before age 25 should be carefully examined for genetic mutations. Hereditary patterns and coexisting features may be useful in determining prognosis.

## 1. Introduction

Juvenile amyotrophic lateral sclerosis (JALS) is a rare neurodegenerative disorder causing degeneration of upper motor neurons (UMN) and lower motor neurons (LMN) [1,2]. JALS is typically defined as onset before the age of 25 years [3]. JALS is associated with considerable disability in the affected young and early death in some forms of the disease. There are several unique differences in JALS compared to adult-onset amyotrophic lateral sclerosis (AO-ALS). First, JALS has a greater genetic contribution, with about 40% of cases related to specific gene mutations [4], while AO-ALS has a genetic contribution of 10% [5,6,7]. The most common genes in the literature associated with JALS are *FUS*, *SETX*, and *ALS2*. *SPG11*, *SOD1*, *SPTLC1*, *UBQNL2*, *SIGMAR1*, and several other genes have also been reported with cases of JALS. The *C9orf72* gene mutation, the most common inherited gene in AO-ALS, has not been reported in JALS. Second, JALS prognosis and disease course varies depending on the gene mutation, with a range from very aggressive to a more indolent, whereas AO-ALS almost uniformly has an aggressive course with death in 2–3 years. Third, JALS cases may have a syndromic presentation, affecting other regions of the central or peripheral nervous system in addition to motor neuron degeneration. 

This is a review of the literature related to JALS and the associated gene mutations. We will be presenting the clinical features, prognosis, electrodiagnostic studies, imaging, and metabolic defects of the main gene mutations associated with JALS.

## 2. Materials and Methods

Literature review through PubMed (Figure 1a) on the following topics: juvenile amyotrophic lateral sclerosis, pediatric amyotrophic lateral sclerosis, and specific gene mutations associated with JALS including *FUS*, *SETX*, *ALS2*, *SOD1*, *SPTLC1*, *SPG11*, *UBQNL2*, *SIGMAR1, GNE, ERLIN1, TARDBP,* and *VRK1*. Individual case reports of genetic associations with JALS were also reviewed. Relevant articles are defined as having JALS in the title or abstract and containing clinical information. The reviewed articles concentrated on genetic associations with JALS, though all JALS articles were included in the initial searches. JALS mutations were not reported in healthy controls. The PubMed search for JALS retrieved a total of 382 articles with 68 relevant articles. There were 9 articles published before 1999 that did not contain any genetic information, though 1–2 of these articles were associated with later papers on JALS genetics. The PubMed search for pediatric ALS retrieved 253 articles, though only 10 were relevant for JALS. An additional PubMed search for gene-specific JALS was also performed, with relevant articles cited in the results section. The EMBASE search of JALS (Figure 1b) resulted in 136 articles, with 69 JALS relevant articles. In EMBASE, there were 28 articles from 1968–1999 on JALS which did not contain information on genetics. EMBASE identified three case reports with novel gene mutations associated with JALS; *SYNE1*, *BICD2*, and *DDHD1*.

## 3. Results

### 3.1. Most Common Gene Associations with JALS

#### 3.1.1. *FUS*

Fused in Sarcoma (*FUS*) gene mutations are most commonly associated with JALS [8,9]. A review of 38 published cases of FUS-JAL [3] showed that the majority of the FUS-JALS are de novo mutations. The median age of onset is 21 years, which places FUS-JALS at the upper limit of the definition for JALS. Clinical presentation of FUS-JALS includes a LMN and UMN presentation of muscle wasting and weakness combined with spasticity and hyperreflexia [10]. FUS-JALS has a rapidly deteriorating course with death due to respiratory muscle failure occurring within 1–2 years. Though bulbar onset is typically associated with a more rapid deterioration, there is no significant difference in survival between spinal onset and bulbar onset ALS in the younger population [11]. Intellectual deterioration has been reported in a few cases of FUS-ALS, including frontal lobe atrophy and hypoperfusion on functional Magnetic Resonance Imaging (fMRI) imaging [12,13]. Movement disorders including myoclonic jerks [14], tremors [15,16], and a case with dysphonia and diplopia [17] have been associated with FUS-JALS. Electrodiagnostic findings are typical of a motor neuronopathy, with evidence of active denervation in the presence of more chronic neurogenic findings and no sensory abnormalities. MRI imaging in one FUS-JALS patient showed increased signals in the pyramidal tracts on T2/FLAIR imaging [12], along with the frontal atrophy mentioned above. MRI of the spine in a second patient showed nerve root enhancement along the spinal cord that did not become suppressed following immunosuppressive treatment [16], which has not been reported in any other case of FUS-ALS. This patient did not have electrodiagnostic or sural nerve evidence of a demyelinating disorder. Sequential computed tomography imaging in this patient showed atrophy of the shoulder girdle and upper limbs, with progressive loss of muscle mass on repeat imaging [12]. Muscle biopsy reports in several FUS-ALS patients showed the expected findings for a motor neuronopathy with fiber-type grouping, target fibers, and muscle fiber necrosis [18]. On electron microscopy, there were autophagic vesicles and lysosomes associated with an autophagic process of motor neuron degeneration. 

Autopsy on four cases of FUS-JALS [19] showed variable corticospinal degeneration and anterior root atrophy. Cytoplasmic basophilic neuronal inclusions are prominent in the Betz cells within the motor cortex, cranial neurons, and anterior horn neurons. Lesser amounts of the cytoplasmic basophilic neuronal inclusions are observed in many other areas of the brain including the substantia nigra, nuclei raphe, inferior olives, and dentate nuclei of the cerebellum. The ultrastructural structure of the inclusions consist of 12–15 nanometer tubulofilamentous structures with electron-dense granules. Immunohistochemistry with anti-FUS antibodies shows localization in both nuclear and cytoplasmic compartments. These findings are specific for FUS-JALS and distinct from sporadic ALS findings of TDP-43 pathology. Similar findings were described in other autopsy studies [20]. 

Mutations associated with FUS-JALS are distinct from AO-ALS even though both have a predilection for the C-terminal nuclear localization signal [11], either as a mutation in the nuclear localization signal or by causing truncation of this region. The propensity for involvement of the nuclear localization signal would suggest that the loss of nuclear FUS may be important to the disease pathology. One study examined 184 FUS-ALS patients, including 34 subjects from a retrospective multicenter review and 150 from published literature, and found that patients with amino acid changes at codon P525L had a median age of onset at 21 years and truncating/frameshift carriers had a median age of onset at 27 years compared to R521 and other mutations [11]. In addition to the younger age of onset, the P525L and truncal shift mutations are associated with a higher frequency of bulbar onset ALS and shorter survival. Several other studies have also shown that P525L patients predominate in JALS and have a short survival. FUS JALS patients frequently present with de novo mutations and no family history [21,22]. 

#### 3.1.2. *SETX*

Senataxin (*SETX*) is the only gene mutation that is typically associated with an autosomal dominant form of JALS named ALS4 [23,24,25]. A series with 31 patients showed a slowly progressive disease course with an average age of onset of 16 years [23]. Initial symptoms included distal lower extremity weakness and difficulty walking. Hand weakness followed by proximal weakness in the upper and lower extremities are later manifestations of ALS4. Bulbar and respiratory systems are generally spared. There is sensory sparing in the majority of cases, with only 7 of 31 patients having clinical signs of distal sensory loss and only one patient with a sensory neuropathy on electrodiagnostic testing. In addition to LMN and UMN findings, one study noted ataxia with dysmetria in the finger-to-nose testing in about half of their patients [23]. Males tend to have greater disease burden and higher serum creatine kinase compared to females. There is also a predisposition for males to be symptomatic (31%) compared to females (13%) [26]. Electrodiagnostic studies show low or absent motor nerve conduction studies with normal sensory nerve conduction studies [24]. Needle EMG studies shows neurogenic changes affecting distal muscles to a greater extent than proximal muscles [23,24,26]. A study of quantitative MRI, which measured fat replacement in skeletal muscle, showed muscle mass loss with diffuse fatty infiltration that is most evident in the distal lower extremities [23]. Neuropathology demonstrates atrophy of both ventral and dorsal roots, with loss of the anterior horn neurons and dorsal columns. The greatest loss of anterior horn neurons is in the lumbosacral region of the cord [24].

Senataxin mutations are also associated with ataxia with oculomotor apraxia type 2 (AOA2) [27,28,29]. This autosomal recessive disorder is characterized by cerebellar ataxia, oculomotor apraxia, axonal sensorimotor neuropathy, and elevated serum α-fetoprotein. Senataxin mutations associated with autosomal dominant JALS occur at distinct locations from AOA2 causing mutations. ALS4 mutations tend to involve the first 400 amino acids, although mutations throughout the gene and including the helicase domain have been reported. *SETX* encodes for a 302.8-kD protein that functions as a DNA/RNA helicase involved in the RNA processing and maintenance of gene integrity [30,31,32]. Alterations in SETX protein function may result in disrupted genomic stability through an alteration of Rad51 recruitment [33]. A depletion of three-stranded nucleic acid structures, called R-loops, and the activation of the transforming growth factor-β (TGF-β) pathway has been found in ALS4 patient tissues [23,34].

#### 3.1.3. *ALS2*/Alsin

*ALS2*, which encodes for alsin, is associated with an autosomal recessive form of JALS, which was originally reported in North African [35,36] and Middle Eastern countries [7,37]. In ALS2-JALS, motor neuron degeneration occurs early and generally progresses slowly [38]. A review of 21 cases in the literature showed that the average age of onset is 4.9 years, with a range of 1–20 years [39]. Clinically, there is an early development of UMN signs of spasticity, dysarthria, dysphagia, and facial weakness. Dysarthria can proceed to anarthria within the first decade of life. Pseudobulbar affect has been noted in several large family cohorts, though cognitive decline is not a prominent feature [35,39]. One case report from Turkey had early onset of spasticity at age 22 months, followed by distal weakness. By the second decade, there is tetraparesis, bulbar weakness, and pseudobulbar affect without cognitive decline [37]. ALS2-JALS patients may also develop scoliosis in their second decade [38]. Brain MRI findings are associated with mild cortical atrophy and thinning of the corticospinal tracts [38,39]. One report of a nerve biopsy and muscle biopsy stated that the findings are typical of a lower motor neuron disorder [35]. In addition to JALS, *ALS2* is associated with a spectrum of disorders, including juvenile primary lateral sclerosis [40], infantile ascending hereditary spastic paraplegia [35], and dystonia [41]. A review of *ALS2* cases in 2020 [39] identified 82 cases of *ALS2*, with 53 cases (64.6%) of hereditary spastic paraparesis, 25 cases of JALS (30.5%), and four cases (4.9%) of Juvenile Primary Lateral Sclerosis (JPLS). Despite the distinct clinical entities, there have not been any specified gene mutations that differentiate these disorders or the disease severity [39,42]. 

*ALS2* or Alsin encodes a 180 kD protein containing three guanine nucleotide exchange factor (GEF) domains named the regulator of chromatin condensation-like domain (RCC1-like domain or RLD), guanine-nucleotide exchange factor for the Rho protein (RhoGEF), and vacuolar protein sorting 9 (VPS9) [43]. ALS2 is located on chromosome 2q33-q34 [35,44]. Binding of Alsin with Rab5 and early endosome antigen-1 (EEA1) results in endosome enlargement, and Alsin may also be involved in microtubule assembly, membrane organization, and trafficking in various cells including neurons [45]. Alsin is involved in membrane transport events, potentially linking endocytic processes and actin cytoskeleton remodeling. Alsin mutations, resulting in a loss of these functions, lead to defects in microtubule axonal transport, endocytic processes, and actin cytoskeleton remodeling [46]. Alterations of these processes likely contribute to neuronal dysfunction and disease [44]. Mutations result in loss of function though frameshift, nonsense, and single-nucleotide polymorphisms, and are inherited in an autosomal recessive inheritance pattern [39,47]. Mutations are found in a distributed pattern across the gene with no particular hot-spot region [39].

### 3.2. Rarer Gene Associations with JALS

JALS is associated with several other gene mutations that have limited family pedigrees or isolated case reports.

#### 3.2.1. *SIGMAR1*

The Sigma-1 receptor (*SIGMAR1*) mutation has been associated with JALS in Middle Eastern families with an autosomal recessive pattern [48] and in a Japanese patient [48,49]. In the initial description of six individuals with SIGMAR1-JALS [48], the clinical onset is at age 1–2 years with spasticity and weakness. Hands and forearms are affected initially with eventual complete atrophy of the forearm extensors and triceps. There is no reported respiratory or bulbar involvement and cognition is preserved. Electrodiagnostic studies confirm the presence of the LMN disorder with sparing of the sensory nerves. MRI studies of the brain are unremarkable. Adult onset SIGMAR1-ALS can be associated with FTD, though cognitive decline is not observed in SIGMAR1-JALS [49,50]. *SIGMAR1* mutations have also been associated with distal hereditary motor neuropathy (dHMN) [51]. Clinical presentation of dHMN is limited to lower motor neuron wasting in the distal legs without evidence of spasticity or sensory abnormalities. SIGMAR1 is an endoplasmic reticulum protein that acts as a chaperone and prevents the accumulation of misfolded proteins [50], participates in lipid transport and ion channel regulation, and has prominent expression in motor neurons of the brainstem and spinal cord [52]. Mutations have been reported in the transmembrane domain [48] and from frameshift mutations upstream at position 95 (p.L95fs) [49]. 

#### 3.2.2. *SOD1*

Copper/zinc superoxide dismutase-1 (*SOD1*) mutations have been associated with three cases of JALS to date [53,54,55]. SOD1 is an important intracellular antioxidant enzyme capable of metabolizing superoxide radicals into oxygen and hydrogen peroxide. A review of three cases in the literature showed onset in the late second decade or early third decade with a combination of UMN and LMN symptoms. All cases have rapid progression of disease with respiratory distress and two reported deaths in less than 2 years. The cases are believed to be de novo mutations with no definitive familial association identified. No sensory symptoms or cognitive decline are noted in the SOD1-JALS cases. An electrodiagnostic study showed needle EMG findings consistent with active denervation and chronic neurogenic changes. Sensory nerve conduction studies are normal. Neuropathological findings in one patient [55] showed prominent anterior horn degeneration and the presence of both Bunina bodies and gliosis in the spinal cord and brain. Anterior horn neurons contain ubiquinated inclusions and SOD1-immunoreactive inclusions. In contrast, the majority of SOD1 AO-ALS cases are autosomal dominant rather than sporadic and have a mean age of onset in the fifth decade [56]. SOD1-JALS mutations occur at or near regions of zinc–ligand binding [54,55] or within the β-strand domain [53], and in most cases are distinct from those found in AO-ALS.

#### 3.2.3. *SPTLC1*

Serine palmitoyltransferase, long-chain base subunit 1 (*SPTLC1*) gene mutations have been associated with JALS [57,58]. SPTLC1-JALS onset is within the first and second decades (range of 4–15). The initial clinical presentation is early onset spasticity and toe walking, followed by LMN degeneration including bulbar involvement. Patients have electrodiagnostic features of active denervation with chronic neurogenic changes, though one case had sensory nerve involvement. Muscle biopsies in four patients showed neurogenic changes consistent with motor neuron disease [58]. 

*SPTLC1* gene mutations are also associated with hereditary sensory autonomic neuropathy (HSAN), with features of small fiber sensory and autonomic dysfunction clinically distinct from JALS. HSAN is associated with elevated levels of deoxysphingolipids levels [59], which is not detected in SPTLC1-JALS patients. SPTLC1-JALS patients are found to have elevated serum levels of ceramide [58]. *SPTLC1* encodes for an essential subunit of serine palmitoyltransferase (SPT), the enzyme that catalyzes the first and rate-limiting step in the de novo synthesis of sphingolipids. *SPTLC1* gene mutations are de novo in most patients, with an autosomal dominant inheritance in one family, and tend to cluster near the first transmembrane domain of exon 2, although mutation in exon 11 was reported in one patient [57,58]. Treatment with serine was initiated in one patient and resulted in weight gain, [57] although the utility of this approach is uncertain and may potentially worsen the overproduction of spingolipids [58]. 

#### 3.2.4. *Spatacsin* (SPG11)

Spastic paraplegia 11 (SPG11), due to mixed heterozygous or homozygous mutations of the *spatacsin* gene on chromosome 15q15–21, has been associated with JALS in two studies [59,60]. Spatacsin may function in the maintenance of cytoskeleton stability and regulation of synaptic vesicle transport. The age of onset for SPG11-JALS ranges from 7 to 23 years, with onset most commonly in the second decade and a disease duration of 34.3 years (range of 27–40). The clinical presentation begins with distal LMN upper or lower extremity weakness combined with UMN signs. Early presentation of bulbar involvement is common in SPG11-JALS, while cognitive deficits and mental health issues are uncommon. MRI of the brain did not show any significant atrophy or defects in the corpus callosum. Autopsy in one individual showed loss of Betz cells, cranial motor loss, and anterior horn neuronal loss. *Spatacsin* gene mutations are also associated with hereditary spastic paraplegia (HSP). However, the features found in HSP patients with progressive UMN signs, sensory neuropathy, and an absence of bulbar involvement distinguish this disease from the SPG11-JALS clinical presentation [61]. Additionally, MRI imaging of HSP shows a thinning of the corpus callosum that is not found in SPG11-JALS. SPG11-JALS mutations typically result in truncation of the protein, with no particular hot-spot region identified [60]. 

#### 3.2.5. *UBQLN2*

Ubiquitin-like protein, specifically ubiquilin 2 (*UBQLN2*), mutations have been associated with X-linked dominant ALS and ALS/dementia in a multigenerational study involving five families with a rare occurrence of JALS [62]. Onset of clinical manifestations of UBQLN2-JALS range from 16 to 71 with the mean onset in males of 33.9 ± 14.0 years and females of 47.3 ± 10.8 years. The disease duration averages four decades, suggesting a slowly progressive disorder. Frontotemporal dementia (FTD) is the most common type of dementia associated with *UBQLN2*. Of the 40 individuals with UBQLN2 mutations, three individuals had onset of symptoms prior to age 24; one with classical ALS findings, one with ALS combined with FTD, and one described as having a combination of upper motor neuron signs and dementia. Pathology of two spinal cords showed deterioration of anterior horn neurons, corticospinal tract atrophy, and astrocytosis. *UBQLN2* encodes for the ubiquitin-like protein ubiquilin 2, which is involved in the regulation of the breakdown of ubiquinated proteins. Mutations are inherited in an X-linked dominant fashion [63] and cluster within the PXX domain of the protein [62]. Defects in *UBQLN2* lead to abnormal accumulation of proteins and neurodegeneration in neuronal cells.

#### 3.2.6. *ERLIN1*

Endoplasmic reticulum lipid raft-associated protein 1 (*ERLIN1*) homozygous gene mutations (c.281T>C and p.Val94Ala) are described in a Turkish family with a four generation pedigree of ALS [64]. Of the 12 affected family members, six family members had onset prior to age 24 years. The typical clinical presentation in this family is lower extremity weakness and spasticity, followed by bulbar symptoms in a slowly progressive motor neuron disorder, resulting in death due to respiratory failure in the fifth–sixth decade [64]. *ERLIN1* mutations have also been reported in HSP complicated by intellectual disability and aphasia [65]. ERLIN1 is a endoplamic reticulum membrane protein that forms a complex with ERLIN2 and participates in endoplasmic reticulum-associated degradation. 

#### 3.2.7. *GNE*

The glucosamine (UDP-*N*-acetyl)-2-epimerase/*N*-acetylmannosamine kinase (*GNE*) mutation has been identified in a family with homozygous mutations and recessive inheritance of ALS [66]. The GNE protein is an important regulator in the synthesis of sialic acid and cell surface sialylation [67]. In this family, the onset of disease ranged from 12 to 35 years of age (mean 26 years). The disease duration is 9–16 years (mean 13.4 years), which is a more intermediate disease course compared to rapid progressors of 1–2 years or slow progressors with the duration of decades. Of the five affected family members, only one presented at age 12 as JALS. This patient presented with severe lower extremity and paraspinal weakness without bulbar symptoms. *GNE* mutations are also associated with distal myopathies with distal weakness but longer duration of disease and myopathic findings on electrodiagnostic studies [67]. The *GNE* mutation in this family with JALS occurred within the ManNAc kinase domain. 

#### 3.2.8. *TARDBP*

The *TAR* DNA binding protein *(TARDBP)* due to a p.Gly348Val heterozygous mutation has only been described in one case of JALS, although mutations in this gene are a well-known cause of AO-ALS [1]. Mutation at this locus with a different amino acid substitution (pGly348Cys) has been detected in AO-ALS. TARDBP binds to nucleic acids and functions in RNA processing and metabolism. The age of onset was age 24 years with initial presentation of distal arm weakness with hyperreflexia in the lower extremity and eventual bulbar symptoms. Death occurred within four years due to respiratory failure. Two other family members had ALS symptoms with a later age of onset but they were not genetically evaluated. In AO-ALS, TARDBP has a mean onset in the sixth decade [68]. 

#### 3.2.9. *VRK1*

The vaccinia-related kinase 1 gene (*VRK1*) mutation causing JALS was initially reported in a 24-year-old Japanese patient who developed intellectual deterioration at age 2 [69]. At age 7, he began to develop gait problems and spasticity. At age 23, he had severe lower extremity weakness, spasticity, and early features of hand weakness. Electrodiagnostic findings showed a motor neuronopathy associated with sensory neuropathy. MRI of the brain showed no atrophy. No other family members were affected with ALS-like symptoms. There is a second case report of two Portuguese patients with initial presentation of distal limb weakness and spasticity [70]. *VRK1* mutations have also been associated with an atypical form of infantile spinal muscular dystrophy with only lower motor neuron findings and pontocerebellar atrophy [71]. JALS cases causing mutations in *VRK1* are inherited in an autosomal recessive or compound heterozygous fashion, and are not detected within a specific hot-spot region.

#### 3.2.10. *SYNE1*

Spectrin repeat containing the nuclear envelope protein 1 gene (*SYNE1*) is associated with JALS in two case reports [72,73]. The SYNE1 protein has a structural role in linking the nuclear plasma membrane with the actin cytoskeleton. In the first case report [72,73], a Japanese male presented with onset of symptoms at age 12 years and was described as having a motor neuron disorder with spastic paraplegia, pes cavus, and dysarthria at age 15. Cognitive decline and limb ataxia were observed at age 30 years. The patient died at age 39 from respiratory compromise. Whole-exome sequencing identified a compound heterozygous mutation inherited from both parents. The second case report [73] describes two Indian siblings with onset of distal wasting and hyperreflexia at ages 11 and 12 with homozygous *SYNE1* deletion mutations. Both case reports noted mild cerebellar atrophy on MRI imaging. *SYNE1* mutations were originally associated with pure cerebellar ataxia [74]. However, in the JALS patients, motor degeneration was more prominent than the cerebellar involvement. SYNE1-JALS cases result from nonsense and frameshift mutations with resulting premature termination of the protein. 

#### 3.2.11. *BICD2*

The BICD cargo adaptor 2 gene (*BICD2*) is associated with one case report of JALS [75]. BICD2 is a motor adaptor protein involved in intracellular transport. A 20-year-old Han woman presented with a combination of UMN and LMN clinical findings and electrodiagnostic evidence of diffuse active and chronic neurogenic findings consistent with a lower motor neuron disorder. She also had dysarthria, mild dysphagia, and pseudobulbar affect. A heterozygous missense mutation was found in the coding region of the *BICD2* gene at c.1196G>A (p.Arg399His). Several studies have reported *BICD2* gene mutations in patients with distal spinal muscular atrophy (SMA) [76]. 

#### 3.2.12. *DDHD1*

DDHD domain containing 1 (*DDHD1*) is associated with one case report of JALS [77]. DDHD1 is an intracellular phospholipase involved in the regulation of mitochondria. In this case, a 16-year-old Chinese male developed combined UMN and LMN clinical findings with electrodiagnostic confirmation of a lower motor neuron disorder. Nerve conduction studies showed both motor and sensory nerve abnormalities despite no symptoms or clinical findings of sensory involvement. Genetic analysis revealed homozygous missense mutations in the *DDHD1* gene at c.1483A>G (p.Met495Val). *DDHD1* mutations have also been associated with the SPG28 subtype of autosomal recessive HSP [78]. 

### 3.3. JALS Mimics

There are a number of disorders that may mimic JALS and, in some cases, may be related to the same genes associated with JALS. Most common ALS mimics are JPLS and HSP, both of which are limited to UMN signs of spasticity and dysarthria. Additionally, HSP may have an earlier and more rapid onset of symptoms, and JPLS may be associated with oculomotor findings [71]. ALS2 is associated with JPLS and HSP. SPG11 and ERLIN1 have been associated with HSP. SMA and dHMN, predominantly LMN disorders, may also mimic JALS forms that have initial LMN presentations. There is one report of a 10-year-old girl with mitochondrial membrane-associated neurodegeneration, a subtype of neurodegeneration with brain iron accumulation (NBIA) who initially had a JALS clinical presentation but later developed iron deposition in the basal ganglia [79]. She had a compound heterozygous mutation in the *C19orf12* gene.

## 4. Discussion

This paper presents 15 gene mutations that have been associated with JALS (Table 1 and Figure 2) The most commonly encountered mutations are in the *FUS* gene. *SETX* and *ALS2* mutations are also frequently found in JALS. JALS cases can be inherited with both autosomal dominant and recessive inheritance patterns. Autosomal recessive inheritance has been observed with *ALS2*, *SIGMAR1, SPG11*, *ERLIN*, *VRK1*, and *GNE* gene mutations. Pathogenic mutations in *UBQNL1* are reported in an X-linked dominant inheritance pattern. De novo mutations were observed in the remaining gene mutations including *FUS* and *SOD1*. Disease severity may limit the likelihood that heterozygous mutations in *FUS* and *SOD1* may be passed on with predicted autosomal dominant inheritance. The age of onset and the rate of disease progression are frequently related to the gene mutation. First decade onset are seen with *ALS2*, *SIGMAR1*, and *VRK1* gene mutations. Other JALS gene mutations have onset in the second or early third decades. The majority of JALS have decade-long slow disease progression. Rapid progression of disease with death in 1–2 years were mainly observed in FUS and SOD1 patients. GNE-JALS and one SYNE1-JALS patient have an intermediate length of disease progression of 1–2 decades. Patients in all 15 gene mutations eventually presented with a clinical combination of UMN and LMN findings, though initial presentations varied by symmetry and bulbar involvement, and predominance was found in upper extremity, lower extremity, or distal limbs. Several gene mutations had additional clinical features: *FUS* with myoclonic jerks and tremor; *SETX* with some cerebellar features; *ALS2* with early anarthria and pseudobulbar affect; and *VRK1* with sensory neuropathy. Cognitive loss is observed in a few cases but is not a major feature of any JALS. Electrodiagnostic studies are useful to confirm LMN involvement but are not useful in distinguishing the different gene mutations associated with JALS. Additionally, many of the genes associated with JALS are also associated with other neurological disorders that may complicate the diagnosis of JALS. Other allelic neurological disorders with the same gene associations are ataxia with oculomotor apraxia (*SETX*), HSP (*ALS2, SPG11, ERLIN1,* and *DDHD1*), JPLS (*ALS2*), dHMN (*SIGMAR1*), distal myopathy (*GNE*), and distal SMA (*BICD2*). Some gene associations with JALS are limited to single case reports and await further confirmation to validate their association with JALS.

In comparison with AO-ALS, JALS had a slower disease progression, with the exception of FUS and SOD1-associated JALS. It is also interesting to note that *FUS* and *SOD1* are among the most common familial ALS gene mutations. However, in JALS, these genes are typically implicated as mainly spontaneous mutations rather than familial gene mutations. Additionally, most of the common familial AO-ALS are inherited in an autosomal dominant fashion, which is not the case for JALS as many cases are de novo. Another difference with AO-ALS is the presence of additional clinical findings, such as movement disorders and cerebellar findings in JALS, that are not typically associated with ALS. It appears that JALS has a greater propensity to involve neurodegeneration in multiple neural pathways while AO-ALS is more limited to motor pathways. To date, there is not a clear explanation for this in the literature but it may be related to the specific mutation loci.

## 5. Conclusions

In conclusion, the genetic testing of patients presenting with a clinical presentation of JALS is important because of the high frequency of gene mutations associated with JALS. Gene identification allows for a better prediction of prognosis and early diagnostic testing in other affected family members. With the future of personalized medicine, the identification of a disease-causing gene mutation may lead to treatment. 

## Figures and Tables

**Figure 1 genes-12-01935-f001:**
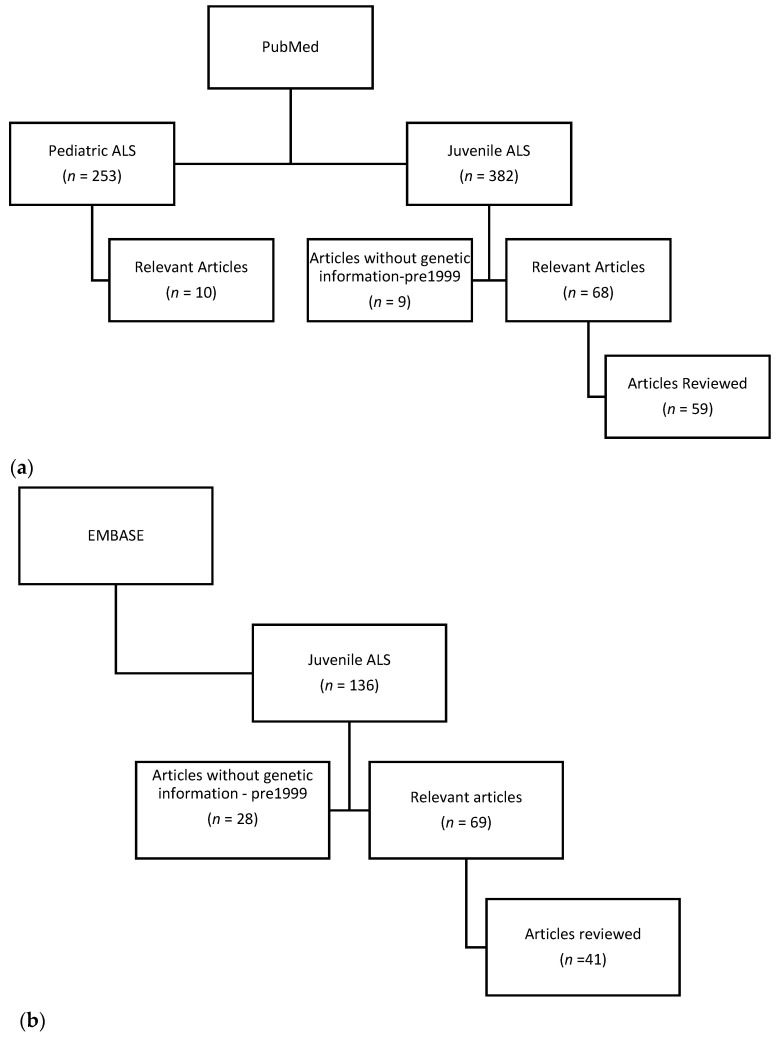
(**a**) Literature search in PubMed. (**b**) Literature search in EMBASE.

**Figure 2 genes-12-01935-f002:**
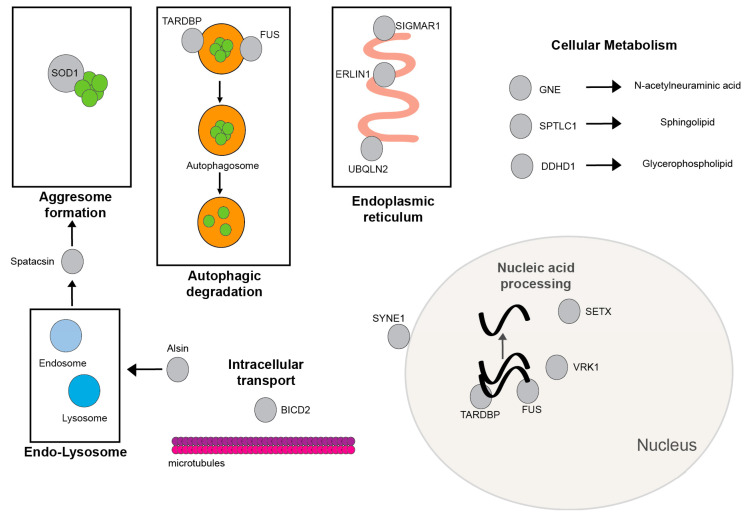
Function and localization of proteins from genes with JALS-associated mutations. Figure legend: Function and localization of proteins from genes with JALS-associated mutations. Proteins are represented as gray circles. GNE, SPTLC1, and DDHD1 function in the metabolism of N-acetylneuraminic acid, sphingolipid, and glycerophospholipid, respectively. FUS: fused in sarcoma; SETX: senataxin; SIGMAR1: sigma-1 receptor; SOD1: copper/zinc superoxide dismutase-1; SPTLC1: serine palmitoyltransferase; long-chain base subunit 1; UBQNL2: ubiquilin 2; VRK1: vaccinia-related kinase 1; ERLIN1: endoplasmic reticulum lipid raft-associated protein 1; GNE: glucosamine (UDP-*N*-acetyl)-2-epimerase; TARDBP: TAR DNA binding protein; SYNE1: spectrin repeat containing nuclear envelope protein 1 gene; BICD2: BICD cargo adaptor 2 gene; and DDHD1: DDHD domain containing 1.

**Table 1 genes-12-01935-t001:** Clinical Presentations of JALS.

**GENE Mutation**	* **FUS** *		* **SETX** *	* **ALS2** *	* **SIGMAR1** *		* **SOD1** *	* **SPTLC1** *	* **SPATACSIN** *
Inheritance pattern	De nova		AD	AR	AR		De nova	De nova and AD	AR
Coding effect	Het.		Het.	Hom. or complex Het.	Hom.		Het.	Het.	Hom. or complex Het.
Age of onset	Early 3rd decade		2nd decade	Early 1st decade	Early 1st decade		Late 2nd decade	1st–2nd decade	2nd–3rd decade
Initial presentation	Bulbar onset		Distal leg weakness	Spasticity, dysarthria, dysphagia & bulbar weak	Spasticity & weakness in forearms		Asymmetric, early LMN signs	Spasticity, then LMN & bulbar involvement	Spasticity, distal weakness & bulbar involvement
Clinical progression	Rapid		Slow	Slow	Slow		Rapid	Slow	Slow
Other features	Myoclonic jerks, tremor		Cerebellar findings	Early anarthria, scoliosis	No bulbar or respiratory symptoms		None noted	Weight loss	None
Cognitive loss	Occasional		None	Pseudobulbar affect, no cognitive decline	None		None	None	None
EDX	Motor neuronopathy		Motor neuronopathy	Motor neuronopathy	Motor neuronopathy		Motor neuronopathy	Motor neuronopathy, sensory changes in one patient	Motor neuronopathy
MRI	Frontal cortical atrophy, pyramidal tract signal		NA	Mild cortical & spinal atrophy	Normal		Normal	NA	Normal
Other non-ALS associated disorders	None		Ocular Apraxia Type 2	JPLS, HSP, dystonia	dHMN		None	HSAN	HSP
Selected references	Refs. [3,8,9]		Refs. [23,24,25]	Refs. [37,38,39]	Refs. [48,49,50]		Refs. [53,54,55]	Refs. [57,58]	Refs. [59,60]
**GENE Mutation**	** *UBQNL2* **	** *ERLIN1* **	** *GNE* **	** *TARDBP* **	** *VRI1* **		** *SYNE1* **	** *BICD2* **	** *DDHD1* **
Inheritance pattern	X-linked dominant	AR	AR	-	AR		AR	De nova	AR
Coding effect	X-linked dominant	Hom.	Hom.	Het.	Hom. or complex Het.		Hom. or complex Het.	Het.	Hom.
Age of onset	2nd decade	2nd decade	2nd–3rd decade	Early 3rd decade	1st decade		2nd decade	2nd decade	2nd decade
Initial presentation	Details NA	Spasticity	Distal lower limb paraspinal, bulbar weakness	Distal upper limb weakness	Lower limb weakness & spasticity, later hand involvement.		Distal predominant weakness	Predominant UMN signs with tongue fasciculations	Details NA
Clinical progression	Slow	Slow	Moderate	Rapid	Slow		Moderate	NA	NA
Other features	None	None	None	None	Sensory		Dysarthria, dysphagia, mild limb ataxia	Dysarthria, dysphagia	None
Cognitive loss	Dementia/FTD	NA	NA	None	Mild	Cognitive decline	Cognitive decline	Pseudobulbar affect	NA
EDX	Motor neuronopathy	Motor neuronopathy	Motor neuronopathy	Motor neuronopathy	Motor neuronopathy		Motor neuronopathy & sensory neuropathy	Motor neuronopathy	Motor neuronopathy & sensory neuropathy
MRI	NA	NA	NA	NA	Normal		Mild cerebellar atrophy	Normal	Normal
Other non-ALS associated disorders	FTD	HSP	Myopathy	None	Infantile SMA		Pure cerebellar ataxia	Distal SMA	HSP
Selected references	Ref. [62]	Ref. [64]	Ref. [66]	Ref. [1]	Refs. [69,70]		Refs. [72,73]	Ref. [75]	Ref. [76]

Abbreviations for Table 1. *FUS*: fused in sarcoma; *SETX*: senataxin; *ALS2*: amyotrophic lateral sclerosis 2; *SIGMAR1*: Sigma-1 receptor; *SOD1*: copper/zinc superoxide dismutase-1; *SPTLC1*: serine palmitoyltransferase; long-chain base subunit 1;; *UBQNL2*: Ubiquilin2; *VRK1*: Vaccinia-related kinase 1; *ERLIN1*: endoplasmic reticulum lipid raft-associated protein 1; *GNE*: glucosamine (UDP-*N*-acetyl)-2-epimerase; *TARDBP*: *TAR* DNA binding protein; *SYNE1*: Spectrin repeat containing nuclear envelope protein 1 gene; *BICD2*: BICD cargo adaptor 2 gene; *DDHD1*: DDHD domain containing 1; LMN: lower motor neuron; NA: not available; UMN: upper motor neuron; FTD: frontotemporal dementia; JPLS: juvenile primary lateral sclerosis; HSP: hereditary spastic paraparesis; dHMN: distal hereditary motor neuropathy; HSAN: hereditary sensory autonomic neuropathy; SMA: spinal muscular atrophy; AD: autosomal dominant, AR: Autosomal recessive, Hom: Homozygous, Het: Heterozygous, EDX: electrodiagnostic testing, MRI: magnetic resonance imaging; Ref.: Reference.

## Data Availability

Data in references only.

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
