# Peer review of "Juvenile Amyotrophic Lateral Sclerosis: A Review"

_genes, 2021, doi:10.3390/genes12121935_

Round 1

Reviewer 1 Report

The review summarizes and comments on the main genetic and clinical features on juvenile ALS. The work is comprehensive, clear and absolutely well written. There is a lack in literature regarding this specific topic, for this reason the paper is going to be of great interest for the readers.

Author Response

Thank-you for you for your comments.

Reviewer 2 Report

Tanya Lehky and Christopher Grunseich provided a complete review about juvenile amyotrophic lateral sclerosis. The manuscript could help clinicians in the identification of the most appropriate genetic test to propose to patients depending on their phenotype. Furthermore, it could be even more useful if a more detailed presentation of genes from a genetic point of view is provided. For example, it could be indicated where the mutations are located in each gene, if there is a mutational hot spot and so on. This could help also people in the laboratory to target the genetic test. Describing the genetic mutations, the authors should indicate if the mutations found in JALS are the same reported in AO-ALS and if these variants are reported in the control population.

Other minor points are:

  • SPTLC1 gene in the text is sometimes indicated as "SPLCT1". Please modify it.
  • "Fused in Sarcoma (FUS) gene mutations are most commonly associated with JALS 64 according to studies from Germany and China". I would remove "from Germany and China" since FUS mutations are the most common in all the popoulations.
  • "TARDBP" is the correct name for the gene encoding for TDP-43. Please modify it, sometimes it is called "TARDP"

Finally, in my opinion there are too many numbers in the "materials and methods" section. There is no need to state how many times a gene has been described in each database, a detailed description is already present in each paragraph. 

Author Response

Reviewer 2:

Tanya Lehky and Christopher Grunseich provided a complete review about juvenile amyotrophic lateral sclerosis.

  1. The manuscript could help clinicians in the identification of the most appropriate genetic test to propose to patients depending on their phenotype.

Response:  Additional genetic information for each gene has been added throughout the manuscript. This information should help the clinician determine the most appropriate analysis needed.

  1. Furthermore, it could be even more useful if a more detailed presentation of genes from a genetic point of view is provided. For example, it could be indicated where the mutations are located in each gene, if there is a mutational hot spot and so on. This could help also people in the laboratory to target the genetic test. Describing the genetic mutations, the authors should indicate if the mutations found in JALS are the same reported in AO-ALS and if these variants are reported in the control population.

Response:  We thank the reviewer for this input. We have provided additional information for each gene to indicate if there is a mutational hot-spot and specify where the JALS associated mutations occur. We have provided text in the materials and methods section to indicate that JALS mutations were not reported in healthy controls (line49). For those genes associated with AO-ALS (FUS, SOD1, TARDBP), we have indicated if the JALS-ALS mutations occur at distinct loci.

Other minor points are:

  • SPTLC1 gene in the text is sometimes indicated as "SPLCT1". Please modify it.

Response:  This has been corrected.

  • "Fused in Sarcoma (FUS) gene mutations are most commonly associated with JALS 64 according to studies from Germany and China". I would remove "from Germany and China" since FUS mutations are the most common in all the populations.

Response:  We removed “from Germany and China” from the sentence.

  • "TARDBP" is the correct name for the gene encoding for TDP-43. Please modify it, sometimes it is called "TARDP"

Response:  This was corrected in two places.

Finally, in my opinion there are too many numbers in the "materials and methods" section. There is no need to state how many times a gene has been described in each database, a detailed description is already present in each paragraph. 

Response:  In Methods and Materials section, the sentence that enumerated the individual gene-associated JALS articles was deleted.

Reviewer 3 Report

In this paper the authors conducted a literature review of JALS and gene mutation associated with JALS. This is a pretty nice work, taking into account that, although rare, this entity is probably underdiagnosed and a systematic description of all cases is very useful for clinical neurologists.

I have only some minor suggestions:

- Line 44: “Relevant articles are defined as having JALS in the title or abstract and containing clinical information”. I was wondering if the authors selected only the studies where genetic diagnoses were made (besides the clinical information). If correct, please specify in the text.

- To simplify the text of the methods, it could be useful to provide a flow-chart/figure to show the methods used for including or excluding the selected studies.

- Please always specify the function of the gene and/or the protein in all the paragraphs, also in more common genes reported (TARBP or FUS).

- In the FUS section, if possible, specify what is the most frequent mutation in FUS gene underlying JALS and if it causes particular clinical phenotypes.

- In the line 80: “the second patient shows nerve root enhancement”, I was wandering: was this the only case reported with this unusual finding? Do the authors reporting this case report give an explanation for this unusual clinical feature?

- In lines 114-115: “There is sensory sparing in the majority of cases but sensory involvement has been reported in a few patients”: please specify the reference of these cases. Did these patients perform EMG? Were NCS normal for sensory nerves?

- Line 192: “Copper/Zinc superoxide dismutase-1 (SOD1) mutations have been associated with some cases of JALS”: Instead of "some" cases, please specify the number, maybe followed by "so far" or "to the best of our knowledge"

- In line 203 please change “spontaneous” with “sporadic”.

- Line 367: “Another difference with AO-ALS, is the presence of additional clinical findings such as movement disorders and cerebellar findings in JALS that are not typically associated with ALS”. Can the authors speculate about a possible explanation of this additional findings in JALS, not present in adult ALS? Is there anything in literature?

Author Response

Reviewer 3:

In this paper the authors conducted a literature review of JALS and gene mutation associated with JALS. This is a pretty nice work, taking into account that, although rare, this entity is probably underdiagnosed and a systematic description of all cases is very useful for clinical neurologists.

I have only some minor suggestions:

- Line 44: “Relevant articles are defined as having JALS in the title or abstract and containing clinical information”. I was wondering if the authors selected only the studies where genetic diagnoses were made (besides the clinical information). If correct, please specify in the text.

Response: Our search wasn’t limited to gene-associated JALS, but we concentrated in reviewing the gene-associated articles.  Most articles after 2000 did discuss gene-associations.  The following sentence was added to the Methods and Materials section:  The reviewed articles concentrated on genetic associations with JALS though all JALS articles were included in the initial searches. 

- To simplify the text of the methods, it could be useful to provide a flow-chart/figure to show the methods used for including or excluding the selected studies.

Response:  Figure 1 with the literature search algorithm was added.

- Please always specify the function of the gene and/or the protein in all the paragraphs, also in more common genes reported (TARBP or FUS).

Response:  We have provided additional functional information for each gene.

- In the FUS section, if possible, specify what is the most frequent mutation in FUS gene underlying JALS and if it causes particular clinical phenotypes.

Response:  The sentence on line 105 was amended to present the most common site of mutations in FUS-ALS.  Mutations associated with FUS-JALS are distinct from AO-ALS even though both have a predilection for the C-terminal nuclear localization signal[11], either as a mutation in the nuclear localization signal or by causing truncation of this region.  The propensity for involvement of the nuclear localization signal would suggest that the loss of nuclear FUS may be important to the disease pathology. 

- In the line 80: “the second patient shows nerve root enhancement”, I was wandering: was this the only case reported with this unusual finding? Do the authors reporting this case report give an explanation for this unusual clinical feature?

Response:  This is the only case report with nerve root enhancement and there is no specific explanation in the case report other than did not respond to immunomodulatory treatment.  The sentence was modified as: MRI of the spine in a second patient shows nerve root enhancement along the spinal cord that did not suppress following immunosuppressive treatment[16] which has not been reported in any other case of FUS-ALS. This patient did not have electrodiagnostic or sural nerve evidence of a demyelinating disorder.  

- In lines 114-115: “There is sensory sparing in the majority of cases but sensory involvement has been reported in a few patients”: please specify the reference of these cases. Did these patients perform EMG? Were NCS normal for sensory nerves?

Response:  This sentence was modified to be more specific about clinical and electrodiagnostic findings: There is sensory sparing in the majority of cases with only 7 of 31 patients having clinical signs of distal sensory loss and only one patient with a sensory neuropathy on electrodiagnostic testing

- Line 192: “Copper/Zinc superoxide dismutase-1 (SOD1) mutations have been associated with some cases of JALS”: Instead of "some" cases, please specify the number, maybe followed by "so far" or "to the best of our knowledge"

Response:  This sentence was made more specific:  Copper/Zinc superoxide dismutase-1 (SOD1) mutations have been associated with  three cases of JALS to date[53-55].

- In line 203 please change “spontaneous” with “sporadic”.

Response:  This change was made.

- Line 367: “Another difference with AO-ALS, is the presence of additional clinical findings such as movement disorders and cerebellar findings in JALS that are not typically associated with ALS”. Can the authors speculate about a possible explanation of this additional findings in JALS, not present in adult ALS? Is there anything in literature?

Response:  This is an interesting question.  We did not find anything specific in the literature to explain a more multisystem degeneration in JALS compared to the AO-ALS that is limited to motor neuron pathways.  The following sentences were added:  It appears that JALS has a greater propensity  to involve neurodegeneration in multiple neural pathways while AO-ALS is more limited to motor pathways.  To date, there is not a clear explanation for this in the literature but may be related to the specific mutation loci.

Round 2

Reviewer 2 Report

The authors succesfully replied to all my concerns. The manuscript has been improved and it is now suitable for publication.